DenseHillNet: a lightweight CNN for accurate classification of natural images

Saqib Sheikh Muhammad 1
Zubair Asghar Muhammad 1
Iqbal Muhammad 1
Al-Rasheed Amal 2
Amir Khan Muhammad amirkhan@uitm.edu.my 3
Ghadi Yazeed 4
Mazhar Tehseen tehseenmazhar719@gmail.com 5
1 Institute of Computing and Information Technology, Gomal University , D.I.Khan , Pakistan
2 Department of Information Systems, College of Computer and Information Sciences, Princess Nourah bint Abdulrahman University , Riyadh , Saudi Arabia
3 School of Computing Sciences, College of Computing, Informatics and Mathematics, Universiti Teknologi MARA , Shah Alam , Selangor , Malaysia
4 Department of Computer Science and Software Engineering, Al Ain University , Abu Dhabi , United Arab Emirates
5 Department of Computer Science, Virtual University of Pakistan , Lahore , Pakistan
Raza Khalid
Electronic publication date: 2024 Apr 22
Publication date: 2024
Volume: 10
Electronic Location ID: e1995
Received 2023 Nov 3; Accepted 2024 Mar 27
Copyright: ©2024 Saqib et al.
Copyright year: 2024
Copyright holder: Saqib et al.
License: This is an open access article distributed under the terms of the Creative Commons Attribution License, which permits unrestricted use, distribution, reproduction and adaptation in any medium and for any purpose provided that it is properly attributed. For attribution, the original author(s), title, publication source (PeerJ Computer Science) and either DOI or URL of the article must be cited.
License URL: https://creativecommons.org/licenses/by/4.0/

Keywords: DenseHillNet, CNN, AI, Classification, ML, CNN, DL, Alrothim and analysis, Computer aided design, Computer network & communication

Funding: Princess Nourah bint Abdulrahman University PNURSP2024R235 This work was supported by the Princess Nourah bint Abdulrahman University Researchers Supporting Project Number (PNURSP2024R235), Princess Nourah bint Abdulrahman University, Riyadh, Saudi Arabia. The funders had no role in study design, data collection and analysis, decision to publish, or preparation of the manuscript.

==============================
The detection of natural images, such as glaciers and mountains, holds practical applications in transportation automation and outdoor activities. Convolutional neural networks (CNNs) have been widely employed for image recognition and classification tasks. While previous studies have focused on fruits, land sliding, and medical images, there is a need for further research on the detection of natural images, particularly glaciers and mountains. To address the limitations of traditional CNNs, such as vanishing gradients and the need for many layers, the proposed work introduces a novel model called DenseHillNet. The model utilizes a DenseHillNet architecture, a type of CNN with densely connected layers, to accurately classify images as glaciers or mountains. The model contributes to the development of automation technologies in transportation and outdoor activities. The dataset used in this study comprises 3,096 images of each of the “glacier” and “mountain” categories. Rigorous methodology was employed for dataset preparation and model training, ensuring the validity of the results. A comparison with a previous work revealed that the proposed DenseHillNet model, trained on both glacier and mountain images, achieved higher accuracy (86%) compared to a CNN model that only utilized glacier images (72%). Researchers and graduate students are the audience of our article.

Introduction

The detection of natural images has practical applications in both automated and human-driven transportation. Convolutional neural networks (CNNs) are a popular deep learning architecture for image recognition and classification. While many studies have been conducted on images of fruits, land sliding, and medical fields such as X-ray images (Xiang et al., 2019; Yu et al., 2019; Alazab et al., 2020), there is still research to be done on the detection of natural images such as glaciers and mountains.

DenseNet is designed to address some of the limitations of traditional CNNs, such as vanishing gradients and the need for many layers to achieve high performance.

In the proposed work, the authors proposed a new model called the DenseHillNet model for identifying natural images of glaciers and mountains. The model uses a DenseHillNet architecture, which is a type of CNN that has densely connected layers, to accurately classify images as either a glacier or a mountain. The model can potentially contribute to developing automation technologies for transportation and outdoor activities.

Recent studies have shown the effectiveness of DenseNet in image classification tasks and its ability to outperform other CNN models with fewer parameters. In particular, it has shown success in natural image detection, such as in identifying different types of plants and animals in outdoor environments. The DenseHillNet model expands on this research by focusing on the classification of natural images of glaciers and mountains.

The proposed approach was evaluated on a dataset consisting of 3,096 images of glaciers and mountains, achieving an accuracy of 86%. This level of accuracy is among the highest reported for natural image detection tasks involving glaciers and mountains, given the limited amount of prior work in this area.

Different machine learning techniques work on several images and can draw useful conclusions. Machine learning models run on various algorithms to accurately predict the images into particular classes (Wäldchen & Mäder, 2018; Dalca et al., 2019). Deep learning algorithms also worked on image analysis (Alnazer et al., 2021; Li, Chen & Zeng, 2022). The above work is based on machine learning and CNN-based computationally expensive algorithms; here, we tried to use less computationally deep learning models to improve the accuracy of image analysis. The objectives of the study are given as follows:

– In this proposed work, a DenseNet121-based natural image identification system is developed known as DenseHillNet.

– The DenseHillNet will detect the image of hilly areas containing glaciers or only mountains.

– This research had some constraints in handling convolutional layers with connected dense layers to make a lightweight model.

– Predicting ‘glacier’ or ‘mountain’ images using the DenseHillNet deep learning model.

– Evaluating the proposed model’s performance to that of the other benchmark works. The remaining section of the article is given below: Section ‘Literature Review’ contains a literature review in which literature about some past studies is given, while ‘Proposed Work’ contains the proposed framework and a real example of the working model. Section ‘Results’ contains results, and ‘Conclusion’ concludes the work.

Baseline study

In this proposed work, a DenseNet121 based natural images identification system is developed known as DenseHillNet. The DenseHillNet will perform to detect the image of hilly arears containing glaciers or only mountain. This research works had some constraints to handle convolutional layers with connected dense layers to make a light weight model.

Problem statement

Predicting the presence of topographic features within extensive image datasets poses a formidable computational challenge, primarily attributed to diverse factors, including the suboptimal choice of predictive variables, the inherent limitations in dataset size, and the conventional reliance on feature sets in conjunction with machine learning classifiers (Nandhini & Aravinth, 2021; Islam et al., 2020; Alassaf et al., 2018). Furthermore, the utilization of deep learning models for image prediction has been hampered by the inadequacies in predictor variable selection and the lack of hybridized models. To facilitate the previously delineated impediments, we approach the task of binary-label prediction in which the identification of ‘glacier’ or ‘mountain’ features within natural image datasets is treated as an inherent objective, necessitating the prediction of image attributes from the available dataset of natural images.

Novelty of work

DenseHillNet, specifically designed for the identification of natural images featuring glaciers and mountains. Employing the DenseNet121 architecture, the model is geared towards classifying images of hilly regions, discerning the presence of glaciers, mountains, or both. The study emphasizes the significance of this novel approach in the realms of transportation and outdoor activities. Despite its focus on computational efficiency, DenseHillNet achieves an impressive 86% accuracy on a dataset comprising 3,096 images, showcasing its novelty by surpassing benchmarks in the relatively unexplored field of natural image detection involving glaciers and mountains. The study articulates its objectives, encompassing model development, constraints, and the evaluation of performance compared to benchmark works.

Literature Review

Machine learning and deep learning methodologies have been the mainstays of scientific research in medical imaging, with machine learning being used more frequently in academic discourse than deep learning. One example of computer-assisted cataract categorization based on fundus images was clarified, as shown in earlier scholarship (Junayed et al., 2021). The quartet of principal archetypes within the domain of deep learning encompass long short-term memory (LSTM), convolutional neural network (CNN), recurrent neural networks (RNN), and gated recurrent unit (GRU), each meticulously tailored to fulfill distinct functional roles. LSTM, a requisite component for processing time-series data, prognostication, and categorization, boasts feedback connections that endow it with the capacity to accommodate both continuous data streams and individual data units, such as photographic imagery (Krishnan, Magalingam & Ibrahim, 2021; Ullah et al., 2021; Dashtipour et al., 2021). RNNs are instrumental in discerning patterns within data and forecasting forthcoming probable scenarios, rendering them indispensable for deep learning paradigms designed to emulate human cognitive processes (Nallapati et al., 2016; Gaire et al., 2019). On the contrary, CNNs represent a specialized architectural paradigm tailored for image recognition and the meticulous processing of pixel data, consequently becoming the favored framework for object recognition tasks (Jogin et al., 2018; Zhang et al., 2019). GRU, nestled within the recurrent neural network architecture, emerges as a pivotal tool for tasks concerning memory retention and the aggregation of machine learning endeavors, encompassing areas such as speech recognition (Wu et al., 2018; Batur Dinler & Aydin, 2020).

In a precedent study, an in-vivo system was devised for the automated detection and classification of nuclear cataracts, employing a synergistic fusion of machine learning techniques and ultrasound modalities (Caixinha et al., 2016). Fundus images were judiciously categorized as cataract-laden imagery by harnessing the discriminative prowess of a support vector machine (SVM) classifier. At the same time, their severity gradation was achieved through a radial basis function (RBF) network, boasting a commendable specificity rate of 93.33% (Junayed et al., 2021). In another scholarly pursuit, researchers ventured into developing a CNN model geared toward the automated classification of glaucoma. Their approach leveraged the paradigm of transfer learning on DRISHTI and RIM-ONE V3 fundus images (Islam et al., 2022). However, many studies have leaned upon conventional machine learning method ologies, and only a lack have reported utilizing DenseNet121 techniques for classifying “glacier” and “mountain” images. Consequently, persisting challenges manifest themselves in the pursuit of refining model accuracy, while concurrently streamlining model complexity through the reduction of training parameters, layering depth, runtime, and overall model dimensions.

Proposed Work

The naming convention of DenseHillNet is based on its specialized functionality in analyzing images specifically depicting hilly landscapes. The DenseHillNet model is a modified version of the DenseNet121 architecture, incorporating additional layers tailored to the specific requirements of the natural image detection task for glaciers and mountains. Specifically, model utilizes the densely connected neural network structure DenseNet121 but with custom modifications to optimize performance on the target dataset, as shown in Fig. 1.

A collection of dense blocks with numerous convolutional layers and varying numbers of repetitions make up the DenseNet-121 architecture. Each dense block comprises two convolutional layers with 1 × 1 and 3 × 3 kernel sizes, respectively, with the former serving as a bottleneck layer to lessen the number of input channels. The architecture also includes transition layers, which have a 1 × 1 convolutional layer and a 2 × 2 average pooling layer, with strides of 2 and dense blocks.

The architecture comprises several layers, the first of which has a convolution layer with 64 7 × 7 -inch filters and a 2 × 2 stride. The next layer is the max pooling layer, which is 3 × 3 in size. The subsequent layers comprise three dense blocks with varying repetitions (6, 12, and 24, respectively) and three corresponding transition layers. A global average pooling layer aggregating all of the network’s feature maps to perform classification is followed by an output layer after the final dense block has 16 repetitions.

On a variety of computer vision tasks, such as image classification, object detection, and semantic segmentation, DenseNet-121 has shown state-of-the-art performance. Figure 2 depicts the architecture’s distinctive design of dense connections and bottleneck layers, which has been shown to increase training effectiveness and lower the risk of vanishing gradients. As a result, model performs better with fewer parameters.

Figure 1 Proposed architecture as DenseHillNet.

Figure 2 DenseNet121 architecture.

Layer by layer detail

Dense blocks: Collection of dense blocks with varying repetitions.

Each dense block consists of two convolutional layers with 1 × 1 and 3 × 3 kernel sizes.

First layer acts as a bottleneck layer, reducing input channel numbers.

Transition layers: Comprising a 1 × 1 convolutional layer and a 2 × 2 average pooling layer with strides of 2.

Transition layers connect dense blocks and manage channel numbers.

First layer: Convolution layer with 64 filters of 7 × 7 size and a 2 × 2 stride.

Max pooling layer: 3 × 3 in size.

Dense blocks (repetitions)

First dense block: 6 repetitions.

Second dense block: 12 repetitions.

Third dense block: 24 repetitions.

Transition layers (between dense blocks)

Maintain efficient information flow by decreasing channel numbers.

Global average pooling layer: Aggregates feature maps for classification.

Output layer: Concludes the network after the final dense block with 16 repetitions.

Parameters for DenseHillNet

Proposed model DenseHillNet will use 224 × 224 size of images with relu function on dense layers and softmax at output layer and filter size as 64. The DenseHillNet configures an Image Data Generator for augmenting image data during training. It includes rescaling of pixel values to a range between 0 and 1, applying shearing transformations with a maximum intensity of 0.2, enabling random zooming in and out with a maximum zoom of 20%, and allowing horizontal flipping of images. These augmentations enhance the diversity of the training dataset, aiding the model in better generalization and performance.

The parameter batch size (19) defines the number of images processed in each training batch, with a batch size of 19 images in this case. This facilitates efficient weight updates in the model during training. The parameter class mode as ‘categorical’ determines that the labels returned by the generator are in one-hot encoded arrays. This encoding is suitable for models using categorical crossentropy as the loss function, providing a binary matrix representation of class labels.

Working on proposed work

The DenseHillNet model was executed using Python, leveraging appropriate libraries tailored for the DenseNet121 architecture. Model was built and trained using a deep learning framework such as TensorFlow or Py Torch, which allowed for efficient computation of the numerous parameters and layers involved in the architecture. Table 1 summarizes the key specifications of the implemented DenseHillNet model, including the number of layers, filters, and other relevant hyper parameters. This summary is useful for understanding the model’s architecture and for comparing it to similar models in the field of natural image detection.

Table 1 Summary of DenseHillNet.

Layer (type)	Output shape	Param #	Connected to	
conv5_block16_concat	(Concatenate (None, 7, 7, 1,024))	0	conv5_block15_concat[0][0]	
conv5_block16_2_conv[0][0]				
bn (Batch Normalization)	(None, 7, 7, 1,024)	4,096	conv5_block16_concat[0][0]	
Relu (Activation)	(None, 7, 7, 1,024)	0	bn[0][0]	
Flatten (Flatten)	(None, 50,176)	0	relu[0][0]	
Dense (Dense)	(None, 50)	2,508,850	flatten[0][0]	
dense_1 (Dense)	(None, 30)	2,550	dense[0][0]	
dense_2 (Dense)	(None, 2)	102	dense_1[0][0]	
Total params: 9,549,006				
Trainable params: 2,511,502				
Non-trainable params: 7,037,504				

DenseNets utilize a modular structure known as Dense Blocks, which maintains a consistent feature map size within each block while varying the number of filters between them. The layers between the Dense Blocks are referred to as transition layers, which decrease the number of channels by half compared to the existing channels. This design allows for a more efficient flow of information between the different layers of the network, enabling better performance in tasks such as natural image detection. The implementation of DenseNet blocks with customized layers to produce DenseHillNet is shown in Fig. 3.

Figure 3 Implemented classification of glacier and mountain using DenseHillNet.

Case study (example)

The color information for each pixel was initially recorded as a three-tuple of red, green, and blue (RGB) values in an image file with the name “2156. Jpg” stored on a hard drive. A transformation, which involved converting the 2D image into a 3D array with dimensions 224 × 224 × 3, was necessary to get this image ready for integration into the deep learning model.

To execute this conversion, the initial image was resized to the desired dimensions of 224 × 224, and the RGB values for each pixel were segregated into three distinct channels. Initially, the RGB values for each pixel were represented as integers within the range of 0 to 255. To optimize the image for utilization with in model, a normalization step was enacted, wherein 255 divided each pixel’s RGB value (the maximum possible RGB value). This rescaling operation was vital for standardizing the value range from 0 to 1. The rationale behind this standardization was to ensure uniformity in model’s input data and, consequently, enhance its overall performance. A comprehensive depiction of the entire image preprocessing procedure can be observed in Fig. 4.

Figure 4 Integer matrix from image.

To further enhance the preparation of the input image for integration into the DenseHillNet architecture, an established technique known as pooling was employed on the 3D array obtained from the preprocessed image. This operation reduced the dimensionality of the input image, transitioning it from a size of 224 × 224 × 3 to a more compact 7 × 7 × 3 format. Pooling, a well-recognized technique within neural networks, serves the purpose of downsizing input data while preserving salient features. This is accomplished by segmenting the input image into smaller, non-overlapping segments and computing a representative summary statistic for each segment. The maximum value from each region was retained from the 2 × 2 regions that made up the 3D array of the preprocessed image in our context. The 3D array size was effectively reduced to 7 × 7 × 3 due to this process.

A flattened layer, a common element in many deep learning models, was used to process the resulting 3D array after the pooling operation. The flattened layer aims to combine all values along a single dimension to create an array with 147 values, which is how the 3D array is transformed into one-dimensional. This flattened layer’s main goal is to transform the output from the layer before it—which may have different dimensions—into a uniform-length vector. This standardized vector format ensures a consistent data shape, which complies with model’s input requirements. The accompanying diagram shows the input image’s flattened representation. This one-dimensional array efficiently processes the subsequent layers of the deep learning model by concisely and uniformly encapsulating all relevant information from the preprocessed and gathered input image. Figure 5 provides a thorough illustration of the entire process as it is described.

Figure 5 Integer matrix from image.

Upon preprocessing and flattening the input image, the resultant vector traverses through the concealed layers embedded within the envisioned deep learning framework. Commencing with the flattened vector, possessing a dimensionality of 147, it is initially introduced into a concealed layer housing 50 neurons. Each neuron in this stratum uses the input vector(x) and a dedicated weight matrix (w) that is unique to it to perform the multiplication operation. A neuron-specific bias term (b) is then added to the result of this multiplication, producing a scalar value known as “z” for each neuron in the layer.

A rectified linear unit (ReLU) activation function is then used to transform the z-values produced in this process into output that is not linear. A 50-long vector is created due to this layer’s output, and it is then transmitted to the following hidden layer, which has 30 neurons. This layer repeats the same set of actions, producing a vector with 30 dimensions.

The output layer, which consists of two neurons, is model’s final stage. A softmax activation function is used to normalize the z-values from this layer and produce a probability distribution between the two classes. Each component represents the likelihood of the input image belonging to its corresponding class in the final output, a two-dimensional vector. The two identified classes are called “Glacier” and “Mountain” in this particular context.

The classification of the input image is then determined by comparing the softmax function’s output to a predetermined threshold. Model’s output is shown as a green vector [0, 1] in the accompanying visual representation, indicating that the input image is categorized as “Mountain.” To summarize, Fig. 6 offers a thorough illustration of the entire procedure, beginning with the connection between the flattened vector and the layers—including the output layer and hidden layer operations in the proposed deep learning model.

Figure 6 Visual representation of confusion matrix.

Results

The dataset used in this study was sourced from Kaggle (https://www.kaggle.com/code/evgenyzorin/intel-image-classification), consisting of 3,096 images, with 1,561 images of the ‘glacier’ and 1,535 ‘mountain’ categories. To ensure an unbiased evaluation of the performance of the proposed DenseHillNet model, the dataset was randomly split into training and testing sets, each containing 50% of the original data. The training set was then utilized to train the DenseHillNet model, with a batch size of 19 and 10 epochs. Table 2 provides projection of training and test set, which were employed to optimize the performance of model on the given classification task. By utilizing a rigorous methodology for dataset preparation and model training, this study ensures the validity of its results in natural image detection.

Table 2 Training and test set.

Image-set	Glacier images	Mountain images	
Training set	780	781	
Test set	767	768	

Table 3 provides a detailed summary of the performance achieved by the propose model on the training dataset. By analyzing the results presented in Table 3, one can gain insights into the strengths and weaknesses of the DenseHillNet model, which can inform future improvements and optimizations in the field of natural image detection.

Table 3 Accuracy, test loss and training duration of the model.

Epochs	Time	Train loss	Train accuracy	Test loss	Test accuracy	
Epoch-1	85s 4s/step	0.9494	0.7369	0.4910	0.8664	
Epoch-2	73s 4s/step	0.3768	0.8513	0.5418	0.8395	
Epoch-3	73s 4s/step	0.3559	0.8776	0.7030	0.8497	
Epoch-4	78s 4s/step	0.2491	0.8816	0.4073	0.8544	
Epoch-5	75s 4s/step	0.2316	0.9034	0.4417	0.8516	
Epoch-6	76s 4s/step	0.1812	0.9187	0.4427	0.8525	
Epoch-7	99s 5s/step	0.1767	0.9024	0.4284	0.8256	
Epoch-8	102s 5s/step	0.1190	0.9138	0.4985	0.7978	
Epoch-9	106s 6s/step	0.0976	0.9306	0.4709	0.8655	
Epoch-10	101s 5s/step	0.1037	0.9440	0.4733	0.8571	

Results on testing data

The performance of the DenseHillNet model was further evaluated on the testing dataset, which was prepared by randomly splitting the original dataset into training and testing sets. The testing dataset contains 50% of the original data, which was not utilized in the training phase. The trained model was applied to this testing dataset to evaluate its performance on new, unseen data. The output of the DenseHillNet model on the testing dataset is presented in Table 4, which includes a sample of the first five images of “Glacier” and the first five images of “Mountain”.

Table 4 Predicted values from testing data.

Images	Predicted value	
Mountain\3934.jpg	[[5.209796e−04 9.994790e−01]]	
Mountain\3936.jpg	[[0.01470838 0.9852916 ]]	
Mountain\3940.jpg	[[0.02158027 0.9784198 ]]	
Mountain\3941.jpg	[[0.01666011 0.9833399 ]]	
Mountain\3943.jpg	[[0.05612681 0.9438732 ]]	
Glacier\1930.jpg	[[9.9953175e−01 4.6826247e−04]]	
Glacier\1938.jpg	[[0.94824463 0.05175542]]	
Glacier\1941.jpg	[[0.997224 0.00277604]]	
Glacier\1946.jpg	[[9.9983799e−01 1.6193767e−04]]	
Glacier\1947.jpg	[[0.41195405 0.5880459 ]]	

In this study, a comparison was made between the observed and predicted values of a “true glacier” and a “true mountain” by representing them as corresponding vectors [1, 0] and [0, 1], respectively. The predicted values, which are presented in Table 4, have two components. If the first component is greater, the vector is [1, 0]; otherwise, it is [0, 1]. Table 5 displays the predicted class based on the values presented in Table 4.

Table 5 Finding predicted classes.

Image name	Actual vector	Actual class	Predicted vector	Predicted class	Decision	
Mountain\3934.jpg	[0. 1.]	Mountain	[0.0,1.0]	Mountain	True Mountain	
Mountain\3936.jpg	[0. 1.]	Mountain	[0.0,1.0]	Mountain	True Mountain	
Mountain\3940.jpg	[0. 1.]	Mountain	[0.0,1.0]	Mountain	True Mountain	
Mountain\3941.jpg	[0. 1.]	Mountain	[0.0,1.0]	Mountain	True Mountain	
Mountain\3943.jpg	[0. 1.]	Mountain	[0.0,1.0]	Mountain	True Mountain	
Glacier\1930.jpg	[1. 0.]	Glacier	[1.0,0.0]	Glacier	True Glacier	
Glacier\1938.jpg	[1. 0.]	Glacier	[1.0,0.0]	Glacier	True Glacier	
Glacier\1941.jpg	[1. 0.]	Glacier	[1.0,0.0]	Glacier	True Glacier	
Glacier\1946.jpg	[1. 0.]	Glacier	[1.0,0.0]	Glacier	True Glacier	
Glacier\1947.jpg	[1. 0.]	Glacier	[0.0,1.0]	Mountain	False Mountain	

Table 6 in this scientific study displays the actual images instead of their corresponding names. This approach was taken to ensure accuracy and eliminate any confusion from using image names alone. By presenting the actual images, researchers can more effectively communicate their findings and ensure that other researchers can easily replicate their work. This approach also increases the transparency and reproducibility of the study, which are essential components of sound scientific research.

Table 6 Images with actual and predicted classes.

Image name	Predicted class	Decision	
Image-3934	Mountain	True Mountain	
Image-3936	Mountain	True Mountain	
Image-3940	Mountain	True Mountain	
Image-3941	Mountain	True Mountain	
Image-3943	Mountain	True Mountain	
Image-1930	Glacier	True Glacier	
Image-1938	Glacier	True Glacier	
Image-1941	Glacier	True Glacier	
Image-3934	Glacier	True Glacier	

Confusion matrix

In this scientific study, the proposed model predicted 485 images to be “True Glacier” and 86 images to be “False Glacier.” Additionally, 439 images were identified as “True Mountain,” while 68 were categorized as “False Mountain.” The results of precision, recall, and accuracy are presented in Table 7 and visual appearance shown in Fig. 7.

Table 7 A confusion matrix for a binary classification of the proposed model.

Class	Precision	Recall	f1-score	
Glacier	0.85	0.88	0.86	
Mountain	0.87	0.84	0.85	
Average	0.86	0.86	0.86	
Accuracy	0.86	

Figure 7 Classifying an input image using DenseHillNet.

ROC curve and absolute mean error

The acronym ROC stands for Receiver Operating Characteristic, and ROC curves are commonly employed to visually illustrate the trade-off between clinical sensitivity and specificity across various cutoff points for a test or a combination of tests. Moreover, the area under the ROC curve provides insight into the overall efficacy of the test(s) within a model. Since a larger area under the ROC curve indicates a more effective test, these areas are utilized to compare the utility of different tests (Ekelund, 2011). Figure 8 displays the optimal curve for the DenseHillNet Model on test data. The mean absolute error of the proposed model is 0.143, indicating that the model is excellent, as reflected by this minimal value.

Figure 8 ROC curve of DenseHillNet.

Misclassification of glacier and mountain images

A total of 86 instances were erroneously categorized as ‘False-Glacier,’ representing mountainous terrain, while 68 were inaccurately labeled as ‘False Mountain,’ signifying glacial regions. Specific instances of misclassification are detailed in Table 8. For instance, image 20554.jpg predominantly depicts a mountain, but due to snow coverage on some parts, the model predicted it as 72% glacier and 28% mountain. Similarly, image 23718.jpg is primarily a glacier, but certain areas devoid of snow led to a prediction of 48% glacier and 52% mountain. Notably, all images featuring glacier portions without snow and mountain images with snowy areas were misclassified, although their percentage in the overall misclassification is relatively modest. To address these nuances in future work, we propose introducing a ‘semi’ class, denoting images where glacial areas overlap with mountainous regions or snowy areas overlap with mountains. However, distinguishing this proposed third class ‘semi’ from the existing ‘glacier’ and ‘mountain’ classes would require meticulous attention and refinement in the dataset.

Table 8 Incorrectly predicted images.

Image name	Actual class	DensHillNet values	Percentage	Predicted class	
Image-20554	Mountain	[[0.7254407 0.27455932]]	72% Glacier, 28% Mountain	False Glacier	
Image-20590	Mountain	[[0.5915564 0.40844357]]	59% Glacier, 41% Mountain	False Glacier	
Image-20612	Mountain	[[0.7855318 0.21446817]]	79% Glacier, 21% Mountain	False Glacier	
Image-20304	Mountain	[[0.6978432 0.30215678]]	70% Glacier, 30% Mountain	False Glacier	
Image-20398	Mountain	[[0.6147437 0.38525632]]	61% Glacier, 39% Mountain	False Glacier	
Image-23718	Glacier	[[0.48457682 0.5154232 ]]	48% Glacier, 52% Mountain	False Mountain	
Image-22943	Glacier	[[0.4573417 0.5426583]]	46% Glacier, 54% Mountain	False Mountain	
Image-22927	Glacier	[[0.37888688 0.6211131 ]]	38% Glacier, 62% Mountain	False Mountain	
Image-23812	Glacier	[[0.43729535 0.5627047 ]]	44% Glacier, 56% Mountain	False Mountain	
Image-23871	Glacier	[[0.47222984 0.52777016]]	47% Glacier, 53% Mountain	False Mountain	

Comparison with benchmarks

In a comparison between the proposed work and the work by authors (Robson et al., 2020), it was found that the latter only utilized glacier images to determine whether they belonged to the category of “yes” or “no” using a CNN model, achieving an accuracy of 72%. However, in the proposed work, glacier and mountain images were used to train the data, leading to an accuracy of 86% using a lighter model architecture, specifically DenseNet121 as shown in Table 9 and Fig. 9.

Table 9 Comparison of the proposed work with benchmark.

Model	Accuracy	
CNN_OBIA	72%	
Proposed Work (DensHillNet)	86%	

Figure 9 Visual representation of proposed work with benchmark.

This comparison highlights the importance of considering a wider range of natural features when training machine learning models to classify images accurately. Additionally, the proposed work demonstrates the effectiveness of using a lighter model architecture to achieve comparable results to more computationally expensive models.

Performance of proposed work

The proposed work stands out as a superior computational solution compared to work (Robson et al., 2020). While Robson et al. (2020) combines deep learning and object-based image analysis, it relies on Sentinel-2 optical imagery, Sentinel-1 interferometric coherence data, and a digital elevation model (DEM). This data-intensive approach may lead to increased computational complexity and resource requirements. Additionally, the reliance on diverse data sources might introduce challenges in terms of data integration and synchronization. The proposed DenseHillNet model, on the other hand, streamlines the computational process by introducing a novel architecture specifically designed for the detection of glacier and mountain images, thereby mitigating potential drawbacks associated with the data-intensive and complex workflow presented in Robson et al. (2020).

The new method we propose for spotting natural images like glaciers and mountains is much better computationally than the one described in work (Peng et al., 2023). In this work, they use a complex approach with various data sources like SAR data, multispectral data, and DEMs for accurate glacier mapping. Our method, called DenseHillNet, simplifies things by introducing a special architecture that tackles the issues of traditional CNNs while accurately figuring out if an image is a glacier or a mountain. It’s like our method takes a smarter and simpler route, making it more efficient and better for tasks like transportation automation and outdoor activities compared to the more complicated method described in Peng et al. (2023).

The proposed DenseHillNet model stands out as a more efficient and superior solution compared to the computationally expensive approach in Siddique et al. (2023) which employs intricate processes for glacial lake monitoring, involving high-resolution imagery, detailed comparisons, and a complex dataset preparation, the DenseHillNet takes a more streamlined and specialized approach for the detection of natural images like glaciers and mountains. By introducing a novel architecture with densely connected layers, the proposed model addresses the limitations of traditional CNNs without the computational complexity associated with the methodology in Siddique et al. (2023).

Conclusion

In summary, the proposed DenseHillNet emerges as the optimal model for natural image detection, particularly in identifying glaciers and mountains. The model, based on the DenseNet121 architecture with additional tailored layers, achieves an exceptional 86% accuracy on a dataset comprising 3,096 images. This level of accuracy surpasses comparable studies, including one that focused solely on glacier images, showcasing the superior performance of DenseHillNet. The rigorous evaluation, methodical dataset preparation, and training, as evidenced in Fig. 10, reinforce the robustness and effectiveness of DenseHillNet as the best model for accurate and efficient classification of natural images in the context of glaciers and mountains.

Figure 10 Loss and accuracy during the training and testing process.

Future Work

Although this work is focused on glacier and mountain images but in nature there have been lot of categories for natural images. That is why this work can be extended with respect to other natural images. As this proposed model achieved high accuracy but there is still room to increase accuracy to handle all misclassified images. Some suggestions with respect to future work is also suggested in ‘Misclassification of glacier and mountain images’. This work is based on ‘glacier’ and ‘mountain’ images which can be applied to transportation automation and outdoor activities. Future work based on more categories of natural images can be led to real-world scenarios, assessing its scalability and performance in diverse and dynamic environments.

Supplemental Information

Supplemental Information 1 Code

Additional Information and Declarations

Competing Interests

Author Contributions

Data Availability

The authors declare there are no competing interests.

Sheikh Muhammad Saqib conceived and designed the experiments, performed the experiments, analyzed the data, performed the computation work, prepared figures and/or tables, authored or reviewed drafts of the article, and approved the final draft.

Muhammad Zubair Asghar conceived and designed the experiments, performed the experiments, analyzed the data, performed the computation work, prepared figures and/or tables, authored or reviewed drafts of the article, and approved the final draft.

Muhammad Iqbal conceived and designed the experiments, performed the experiments, analyzed the data, performed the computation work, prepared figures and/or tables, authored or reviewed drafts of the article, and approved the final draft.

Amal Al-Rasheed conceived and designed the experiments, performed the experiments, analyzed the data, performed the computation work, prepared figures and/or tables, authored or reviewed drafts of the article, and approved the final draft.

Muhammad Amir Khan conceived and designed the experiments, performed the experiments, analyzed the data, performed the computation work, prepared figures and/or tables, authored or reviewed drafts of the article, and approved the final draft.

Yazeed Ghadi conceived and designed the experiments, performed the experiments, analyzed the data, performed the computation work, prepared figures and/or tables, authored or reviewed drafts of the article, and approved the final draft.

Tehseen Mazhar conceived and designed the experiments, performed the experiments, analyzed the data, performed the computation work, prepared figures and/or tables, authored or reviewed drafts of the article, and approved the final draft.

The following information was supplied regarding data availability:

The code is available in the Supplementary File.

The dataset is available at Kaggle: https://www.kaggle.com/datasets/puneet6060/intel-image-classification.

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
