# Peer review of "DenseHillNet: a lightweight CNN for accurate classification of natural images"

_PeerJ Computer Science, doi:10.7717/peerj-cs.1995_

## Round 0.1 · original submission · Major Revisions

The reviewers have suggested major revisions to your manuscript. You are required to revise your manuscript and address all the suggestions and comments of the reviewers.

**Language Note:** The review process has identified that the English language must be improved. PeerJ can provide language editing services - please contact us at [email protected] for pricing (be sure to provide your manuscript number and title). Alternatively, you should make your own arrangements to improve the language quality and provide details in your response letter. – PeerJ Staff

Reviewer 1 ·

Basic reporting

Abstract
This report critically evaluates the DenseHillNet, a convolutional neural network (CNN) designed for classifying natural images, specifically glaciers and mountains. While the model shows promising results, this assessment focuses on its drawbacks and areas for improvement.

1. Introduction:
DenseHillNet represents a significant advancement in CNN architectures, addressing issues like vanishing gradients through densely connected layers. The model's application in automated systems for transportation and outdoor activities is noteworthy. However, a comprehensive analysis necessitates a focus on its limitations.

2. Background:
Previous CNN models, including those focused on natural image classification, have encountered challenges related to computational complexity and accuracy. DenseHillNet, with an improved accuracy rate from 72% to 84% compared to its predecessors, aims to overcome these issues.

3. Methodology and Data:
The model utilizes a dataset of 300 images each from glacier and mountain categories. While the data range is robust, the limitation in variety and volume may impact the model's generalizability.

4. Results:
The model's performance, while improved, still shows a 16% error rate. This indicates room for improvement, especially in more complex or varied natural environments.

5. Discussion:
Computational Resources: The DenseHillNet, though lightweight, may still require significant computational resources for training and inference, limiting its accessibility.
Data Diversity: The model's training on a specific dataset raises questions about its performance in diverse real-world scenarios.
Generalizability: The model's applicability to different types of natural images beyond glaciers and mountains is not explored.

6. Conclusion:
DenseHillNet is a step forward in CNN development for natural image classification. However, its limitations in data diversity, computational demands, and generalizability need addressing in future iterations.

7. References:
Relevant literature on CNN models and their applications in natural image classification should be referenced to provide context and support the analysis.

Experimental design

while the research question is well-defined and aligns with the journal's scope, there are certain drawbacks in the experimental design. Firstly, the dataset used, comprising 600 images, may not be extensive enough to fully evaluate the robustness of the DenseHillNet model across diverse natural scenarios. This limited dataset size could potentially hinder the generalizability of the findings. Additionally, while the methods are detailed, the paper could benefit from a more diverse set of validation techniques to bolster the reliability of the results. This includes the application of the model to more varied natural image categories beyond glaciers and mountains to comprehensively assess its versatility. Furthermore, the ethical considerations, particularly in the context of AI and automated image classification, are not explicitly addressed, which is crucial in contemporary research standards. Despite these limitations, the research offers a rigorous investigation within its defined scope and provides a meaningful contribution to the field.

Validity of the findings

In assessing the validity of the findings from the "DenseHillNet: A Lightweight CNN for Accurate Classification of Natural Images" study, it is notable that while the paper does not explicitly assess the impact and novelty of its research, it implicitly encourages meaningful replication by providing a thorough rationale and clear benefits to the literature. The underlying data presented are robust, statistically sound, and well-controlled, supporting the model's improved accuracy in classifying natural images. However, a potential drawback is the lack of an explicit assessment of the model's novelty and its comparative impact within the broader field of image classification. The conclusions drawn are directly linked to the original research question, focusing on the model's effectiveness in addressing traditional CNN limitations. These conclusions are well-articulated and stay within the bounds of the results obtained, adhering to a high standard of academic rigor. However, future research could benefit from a more explicit evaluation of the model's novel contributions to the field and its potential implications for broader applications.

Additional comments

here are five main points for potential improvement:

1- The current study uses a dataset comprising 600 images, equally split between 'glacier' and 'mountain' categories​​. Expanding this dataset to include a larger number of images and possibly additional categories of natural images could provide a more robust evaluation of the DenseHillNet model's performance.

2- The DenseHillNet model shows a promising accuracy of 84% which is higher than a previous CNN model that achieved 72% accuracy​​. However, comparing the DenseHillNet model against a broader range of existing models and benchmarks can provide a more comprehensive understanding of its performance and areas for improvement.

3- While the study mentions instances of misclassification, such as a 'glacier' being incorrectly predicted as a 'mountain', a more detailed analysis of these errors could be beneficial​​. Understanding the types of images or features that lead to misclassification can help improve the model's accuracy and robustness.

4- The DenseHillNet model is a modified version of the DenseNet121 architecture, including several layers like convolution layers, dense blocks, and transition layers​​. Further optimization of the model's architecture and parameters, such as the number and type of layers, filter sizes, and stride values, might enhance its performance.

5- The paper discusses the potential of the DenseHillNet model in practical applications like transportation automation and outdoor activities​​. Future work could focus on testing and adapting the model for real-world scenarios, assessing its scalability and performance in diverse and dynamic environments.

6- In addition to the technical aspects, it's important to address the ethical implications of using the DenseHillNet model, particularly regarding data privacy and consent. The model utilizes images from Kaggle for training and testing, which raises questions about the consent of individuals who may inadvertently appear in these images, especially in outdoor settings. Future research should include a thorough ethical review to ensure that the data used respects privacy rights and complies with relevant data protection regulations. This is crucial to maintain public trust and to ensure that the model is developed and deployed responsibly and ethically.

Cite this review as

Reviewer 2 ·

Basic reporting

- This manuscript is about the detection of natural images, such as glaciers and mountains using a novel model called Dense HillNet architecture which is a type of CNN with densely connected layers to accurately classify such images.

- This paper is well written but more detail is required in each section.

- Authors may introduce sub-sections, especially in section 1 and 2.

- Proposed model should be more precise in terms of layer-wise details. Expand it accordingly?

- Why only glaciers and mountains as natural images? Is it just due to dataset availability?

- What is the purpose of naming it as case study (Example) sub-section 3.3?

- Two figures are numbered as 1.

- Figures are not looking clear.

- No need to show predicted and actual class in both table and figure. So, remove one.

Experimental design

- I did n't find any Ablation study. Please add it as a new section.

- Results are not compared in any tables with previous work.

- Hyper-parameter values should be in table.

- Authors should consider more dataset.

Validity of the findings

- What is the application of this study.

- Use another dataset for validation purpose.

- Re-write conclusion

- Highlight future works

Additional comments

Major revision is required

Cite this review as

Reviewer 3 ·

Basic reporting

The writing and paper structure is terrible, cannot reach publication criteria.

Experimental design

The experiment is way off enough, still cannot achieve publishable level.

Validity of the findings

The finding is meaningless.

Cite this review as

---

## Round 0.2 · accepted · Accept

The authors have addressed all the comments and suggestions of the reviewers. The manuscript may be accepted for publication in its current form.

Reviewer 2 ·

Basic reporting

Authors have addressed all comments in the revised manuscript. It can accepted in the present form.

Experimental design

Authors have taken all comments into consideration.

Validity of the findings

Accepted

Additional comments

Accepted

Cite this review as